# The audio features of sleep music: Universal and subgroup characteristics

**Rebecca Jane Scarratt**[1,2], **Ole Adrian Heggli**[2], **Peter Vuust**[2], **Kira Vibe Jespersen**[2]*

**1** Radboud University, Nijmegen, The Netherlands, **2** Center for Music in the Brain, Department of Clinical Medicine, Aarhus University & The Royal Academy of Music Aarhus/Aalborg, Aarhus, Denmark

* kira@clin.au.dk

**Data Availability Statement:** All relevant data files are available from the GitHub database (https://github.com/RebeccaJaneScarratt/SpotifySleepPlaylists).

## Abstract

Throughout history, lullabies have been used to help children sleep, and today, with the increasing accessibility of recorded music, many people report listening to music as a tool to improve sleep. Nevertheless, we know very little about this common human habit. In this study, we elucidated the characteristics of music associated with sleep by extracting audio features from a large number of tracks (N = 225,626) retrieved from sleep playlists at the global streaming platform Spotify. Compared to music in general, we found that sleep music was softer and slower; it was more often instrumental (i.e. without lyrics) and played on acoustic instruments. Yet, a large amount of variation was present in sleep music, which clustered into six distinct subgroups. Strikingly, three of the subgroups included popular tracks that were faster, louder, and more energetic than average sleep music. The findings reveal previously unknown aspects of the audio features of sleep music and highlight the individual variation in the choice of music used for sleep. By using digital traces, we were able to determine the universal and subgroup characteristics of sleep music in a unique, global dataset, advancing our understanding of how humans use music to regulate their behaviour in everyday life.

## Introduction

Despite sleep being essential for human health and well-being, sleep problems are increasing in modern society [1–3]. Although some people seek professional help for their sleep problems, many people choose to initiate self-help strategies such as listening to music [4–6]. Indeed, epidemiological studies show that up to 46% of respondents indicate that they use music to help themselves fall asleep [6–8] which can significantly improve sleep across adult populations [9–11]. However, it is not well understood what defines the music that people use to sleep. Are there specific universal features characterising music used for sleep? Or may music be used as sleep aid independently of its musical features? In this study, we address these questions using big data from the global streaming service Spotify.

The habit of using music for sleep improvement may be rooted in the ubiquitous propensity of caregivers to sing lullabies to their babies [12, 13]. Lullabies are often sung to babies to assist with falling asleep and research indicates that even unfamiliar lullabies from different cultures

**Funding:** The author(s) received no specific funding for this work.

**Competing interests:** The authors have declared that no competing interests exist.

decrease arousal, heart rate and pupil size in babies [14]. As such, it has been hypothesised that music facilitates sleep by reducing arousal [15–17]. This may be physiologically or psychologically through a pleasurable emotional response, or by acting as a distractor from stressful thoughts. In general, it has been argued that in order to facilitate a relaxation response, music should have simple repetitive rhythms and melodies, small changes in dynamics, slow tempi (around 60-80bpm), no percussive instruments, and minimal vocalisations [18–20]. However, these claims have not been investigated in relation to sleep.

Previous research on sleep music characteristics is limited by the use of qualitative self-reports with relatively small amounts of data, usually in geographically restricted areas. One survey study based in the UK (N = 651) found a large diversity among music used for sleep and concluded that the choice of music was driven more by individual differences than any consistent type of music [17]. However, that study only collected information on artists and genres and did not examine the specific characteristics and audio features of the actual music. Similarly, an Australian survey study on students (N = 161) found that music that aided sleep was characterised by medium tempo, legato articulation, major mode and the presence of lyrics [21]. Because that study was restricted to only 167 pieces of music in a local student population, it is unlikely to represent a full image of the type of music use for sleep. Therefore, a large global sample investigating not only genre but also the audio features of the music is important to understand the characteristics of music used for sleep.

Today, music listening is very often done via international streaming services and this allows for the collection of big data on sleep music from around the globe [22]. In 2019, the International Federation of the Phonographic Industry reported that 89% out of 34 000 internet users listened to music via a streaming service such as Apple Music, Spotify or YouTube music [23]. Out of these services, Spotify stands out with over 320 million listeners worldwide in 2020 [24, 25]. In addition, Spotify offers an easily accessible API (application programming interface), allowing users and researchers to pull metadata and pre-calculated audio features from millions of unique tracks [26, 27].

The audio features available from Spotify describe both basic features of recorded music, such as its tempo and loudness, and compound measures indicative of, for instance, a particular track's Danceability and Acousticness. While the calculations behind these audio features are not publicly available, they nonetheless provide a rich source of perceptually relevant information, in particular for quantifying differences between and within datasets using the same set of audio features. Leveraging these audio features allows us to use Spotify as a platform for investigating sleep associated music in a representative industrialised population [28, 29].

By amalgamating data from Spotify, we build a large database of music associated with sleep, and related metadata and audio features. Using multiple analysis approaches, we use this dataset to determine both universal and subgroup characteristics of sleep music.

## Materials and methods

### Building the Sleep Playlist Dataset

We used Spotify to build a dataset of sleep-associated playlists of musical tracks. We used the playlist search function in the Spotify desktop client, searching for all playlists including a word in the word family of "sleep" (e.g. sleep, sleepy, sleeping) either in the title or in the description. The search also brought up results in different languages such as dormir, dormire, slaap, søve etc. With this inclusive search, we aimed to retrieve all relevant playlists. At the same time, we wanted to ensure that the playlists reflected the use of music for human sleep, and we therefore developed four exclusion criteria: we excluded playlists aimed at dogs or

other pets, non-music playlists (e.g. podcasts, ASMR and nature sounds), and playlists where the word sleep did not refer to the use of music for sleep (e.g. band names and soundtracks including the word 'sleep'). To make sure our dataset was representative of general trends in sleep music, and not just individual idiosyncrasies, we also excluded playlists with less than 100 followers.

The search was performed during the fall of 2020. As the exact number of sleep-related playlists on the Spotify platform is only available with access to Spotify's proprietary database, we stopped data collection at 1,263 playlists. A total of 248 playlists were excluded due to either the criterion of having less than 100 followers or other predetermined exclusion criteria (S1 Table). The title, content and purpose of 69 playlists were ambiguous, so a qualitative review by two of the authors were performed by inspecting playlist title, description, visuals and content. Of these, 29 were excluded, leaving the total number of playlists in our dataset at 986. A flow diagram of the procedure can be found in Fig 1. The data includes no personal data from Spotify users and the data collection complies with Spotify's terms of use [30]. The thorough assessment procedure was aimed to ensure that the included playlists were indeed associated with sleep. While we cannot experimentally ascertain their use, the general descriptions of the playlists, such as "Soothing minimalist ambient for deep sleep" or "A series of soothing sounds to softly send you to sweet, sweet slumber" and the visual illustration accompanying many of the playlists, such as a photo of a bed, a pillow, or a sleeping person, indicates the notion of sleep. Furthermore, all of the cases in which the mention of the word "sleep" was ambiguous were evaluated individually and excluded if any doubt persisted. While there is a possibility that some included playlists were not meant for the sleep related purposes, we believe the size of the dataset reduces their potential impact.

For each playlist, the playlist link, title, description, creator, number of followers, number of tracks and duration was noted. We used Spotify's API through Spotipy in Python to access and extract audio features from the tracks included in the SPD. For one of the playlists, we were unable to access metadata and audio features, leaving the dataset used for further analysis at 985 playlists holding a total of 225,626 tracks. Out of these tracks, 95,476 tracks appeared in multiple playlists, leaving 130,150 unique tracks. In terms of followers, the playlist had a median following of 1,932 users, with a minimum of 102 and a maximum of 3,982,105. The playlists have a median of 434 tracks, with a minimum of 2 and a maximum of 9,991 (S2 Table).

The precalculated audio features available from Spotify cover a wide range of both basic and compound musical measures. Notably, as the calculation of these audio features are proprietary, we are unable to quantify exactly which calculations and transformations underlie each feature. Therefore, we base our interpretation of the audio features on Spotify's description as part of their API reference manual [31], which we provide a summary of in Table 1.

## Genre

Spotify provides a highly detailed genre description of its tracks, with examples such as "Icelandic post-punk" and "instrumental math rock", and occasionally returns multiple of these genres. To provide a more broad view on the genres included in our dataset we applied a genre reduction algorithm [32]. This algorithm aims to reduce the list of sub-genres provided by Spotify for a particular track such that: $G(x) = argmax_y(\sum_{i=1}^{n} g_y(x_i))$, where $x$ is the list of sub-genres of a track, and $G(x)$ is the main genre, obtained by calculating whether each pre-determined main genre $y$ is a substring of the sub-genre $x_i$, and then choosing the main genre with the most occurrences. A Python-implementation is available at GitHub.com/RebeccaJaneScarratt/SpotifySleepPlaylists.

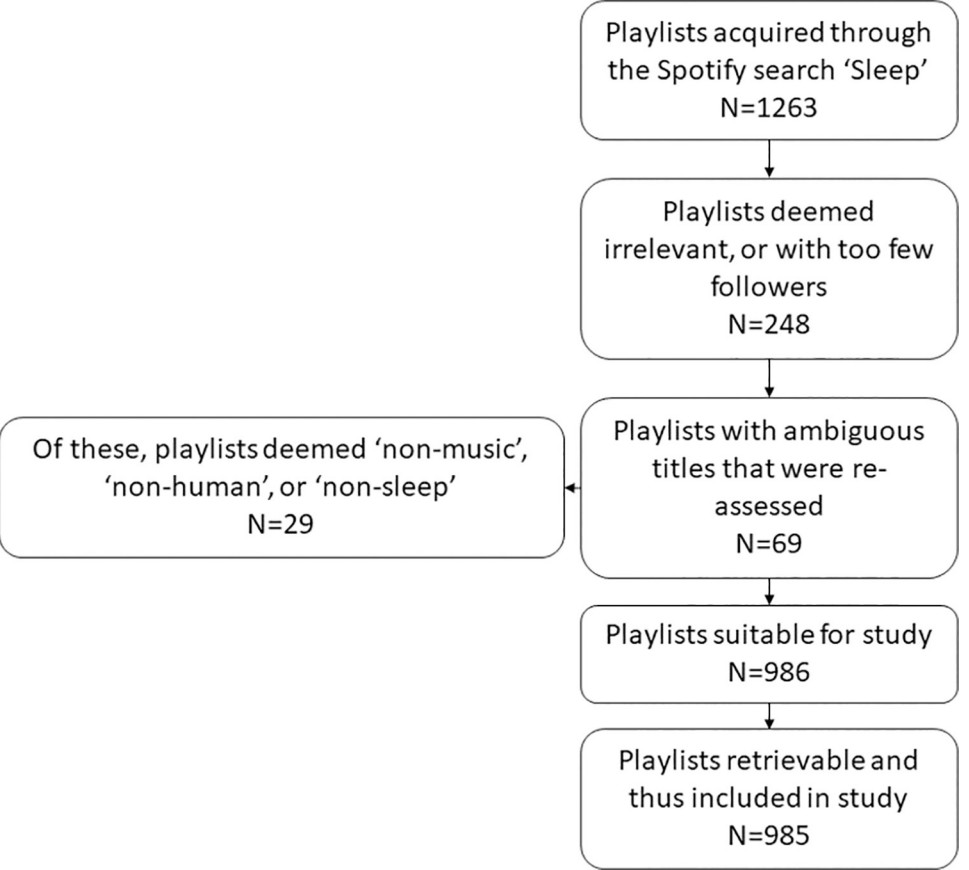

**Fig 1. Flowchart of the playlist search and exclusion procedure.** We acquired 1,263 playlists by searching Spotify for 'sleep' words in title or description. 248 were excluded based on the four exclusion criteria non-human (e.g. music to help your dog sleep), non-music (e.g. speech, ASMR, nature sounds), non-sleep (e.g. sleep as part of a band name or soundtrack), or non-representative (less than 100 followers). 69 playlists had ambiguous titles (such as "NO SLEEP"), which were then qualitatively reviewed, leading to 29 additional exclusions. One playlist had unretrievable metadata and was excluded. The final dataset included 985 playlists.

Our list of main genres was created from the 23 STOMP genres removing Oldies with 5 additional genres that were added in Trahan et al. [17] and adding 4 genres ourselves [33]. For a full overview of the 31 genres, see Table 2.

## Selecting a control dataset

To assess the specific characteristics of sleep music, we selected the Music Streaming Sessions Dataset (MSSD) as a control dataset. This publicly available dataset was released by Spotify on CrowdAI [34] and contains audio features for approximately 3.7 million unique tracks that were listened to at all hours of the day. The MSSD was collected over multiple weeks in 2019 and is treated here as representative of general music listening on Spotify.

## Analyses

To statistically assess the specific characteristics of sleep music, we compared the unique tracks from our Sleep Playlist Dataset (SPD) to the Music Streaming Sessions Dataset (MSSD). First, we statistically compared the individual audio features between sleep and general music using

**Table 1. Overview of the audio features that are accessible through the Spotify API and their descriptions as given by Spotify [31].**

| Audio feature | Description |
|---|---|
| Loudness | A value indicating the overall loudness of a track, ranging between -60 and 0 dB. Spotify does not specify a dB scale, but it is assumed this is measured in LUFS or a similar perceptual loudness scale. |
| Energy | A value indicating a perceptual measure of intensity and activity, ranging between 0 and 1. |
| Acousticness | A confidence value indicating how likely a track is acoustic, meaning performed on non-amplified instruments, ranging between 0 and 1. |
| Instrumentalness | A confidence value indicating how likely a track contains no vocals, ranging between 0 and 1 with values above 0.5 likely to be instrumental tracks. |
| Danceability | A value indicating how suitable a track is for dancing, ranging between 0 and 1, with higher values indicating increased danceability. |
| Valence | A value indicating positively valenced a track is (from a Western point of view), ranging between 0 and 1, with high values indicated a track that is likely to be perceived as more positive, happy, and cheerful |
| Tempo | A value indicating the speed or pace of track, as estimated by the average beat duration, given in beats-per-minute (BPM). |
| Liveness | A confidence value indicating how likely a track is performed live, for instance by detecting the sound of an audience in a recording. |
| Speechiness | A value indicating the presence of spoken words in a track. |

Welch's t-tests from the rstatix package to account for unequal size and variance. All p-values were FDR-corrected. Second, we used linear discriminant analysis (LDA) using the flipMulti-variates package distributed by Displayr to identify the individual audio feature's contribution to separating the two datasets. Due to the unequal size between the SPD and the MSSD, data from the former was weighted by a factor of 28.48.

To assess the degree to which sleep music can be considered one homogeneous group of music or whether different subgroups exist within this category, we used a $k$-means clustering approach. The clustering was performed using R's inbuilt kmeans function, with a maximum of 1000 iterations. This clustering approach partitions the data into $k$ clusters by minimizing the within-cluster variance. Selecting the optimal $k$ depends on the intended outcome of the clustering, with lower values of $k$ generally capturing larger clusters in the data. To determine the optimal $k$ for our case we applied the elbow-method, wherein the within-cluster sum-of-squares is summed for each value of $k$, in our case $k = [1,17]$. Inspecting this value revealed an optimal partition of the dataset into seven clusters.

All statistical analyses were performed in RStudio version 1.3.959 using R version 4.0.0, running on Windows 10. The scripts used for analysing the dataset can be found at GitHub.com/RebeccaJaneScarratt/SpotifySleepPlaylists. Figs were made using ggplot2 and the Rain-CloudPlots package [35].

**Table 2. All genre categories used to reduce the genre tags.**

| Origin | Genres |
|---|---|
| STOMP Genres | Electronic/dance, New age, World, Pop, Country, Christian, Blues, Jazz, Bluegrass, Folk, Classical, Gospel, Opera, Rock, Punk, Alternative, Heavy metal, Rap, R&B, Funk, Reggae, Soundtrack |
| From Trahan et al. (2018) | Ambient, Instrumental, Indie, Meditation, House |
| Additional genres | Sleep, Background, Lo-fi, Lullaby |

## Results

### Defining features of sleep music

The comparison between general music in the MSSD and sleep music in our SPD yielded statistically significant differences for all audio features (p < .001), for both the t-test comparison and for the LDA. To better interpret our results, we focus on effect size, as measured by Cohen's $d$ for the statistical comparison and $r^2$ for the LDA [36]. The results are illustrated in Fig 2.

The largest effect sizes were found for a decrease in Loudness (Cohen's d = -1.25) and Energy (Cohen's d = -1.46) and for an increase in Acousticness (Cohen's d = 1.20) and Instrumentalness (Cohen's d = 1.10) in the SPD compared to the MSSD. Danceability (Cohen's d = -0.64), Valence (Cohen's d = -0.93), Tempo (Cohen's d = -0.47), Liveness (Cohen's d = -0.34), and Speechiness (Cohen's d = -0.35) were all significantly lower in the SPD as compared to the MSSD. For a full overview, see Table 3.

The LDA performed at a correct classification rate of 78.61% (MSSD = 79.86%, SPD = 77.36%), well above chance levels. All audio features were found to significantly contribute to the classification. The best discriminator was Loudness ($r^2$ = .09), followed by Energy ($r^2$ = .06), Acousticness ($r^2$ = .04), Instrumentalness ($r^2$ = .04), Danceability ($r^2$ = .02), Valence ($r^2$ = .02), Tempo ($r^2$ = .01), Liveness ($r^2$ < .01), and Speechiness ($r^2$ < .01).

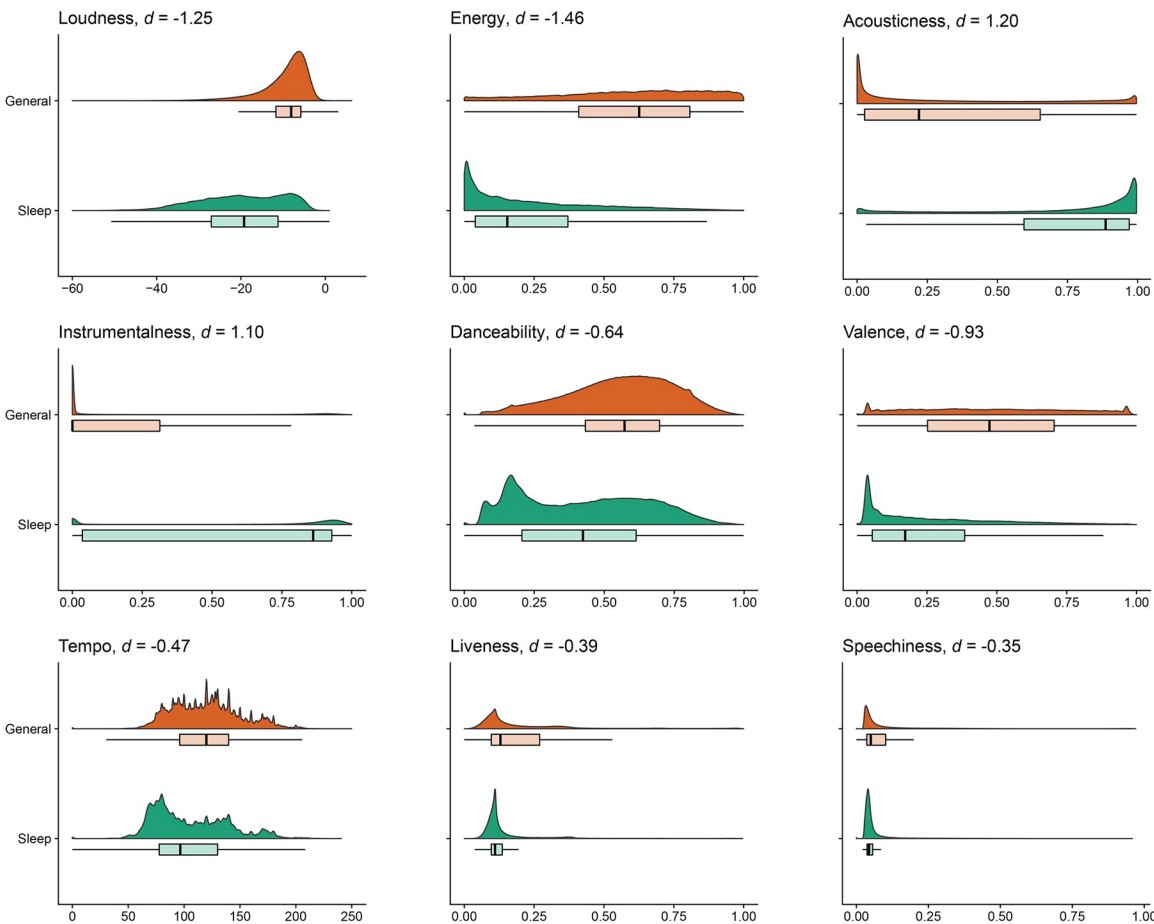

**Fig 2. Audio feature comparisons between sleep music in the SPD (green) and general music in the MSSD (orange).** The panels show the individual audio features, illustrated as smoothed density plots with an underlying box plot wherein the vertical line represents the median value, with the associated Cohen's d for the comparison of sleep music versus general music.

**Table 3. Statistical comparison and linear discriminant analysis of audio feature between the Music Streaming Sessions Dataset and the Sleep Playlist Dataset.**

| Audio feature | Dataset | | Statistical comparison | | LDA | |
|---|---|---|---|---|---|---|
| | MSSD (N = 3,706,388) | SPD (N = 130,150) | Cohen's D | P | R-Squared | P |
| Loudness | -9.6 | -19.78 | -1.25 | < .001 | .09 | < .001 |
| Energy | .59 | .23 | -1.46 | < .001 | .06 | < .001 |
| Acousticness | .35 | .74 | 1.20 | < .001 | .04 | < .001 |
| Instrumentalness | .21 | .62 | 1.10 | < .001 | .04 | < .001 |
| Danceability | .56 | .42 | -0.64 | < .001 | .02 | < .001 |
| Valence | .48 | .25 | -0.93 | < .001 | .02 | < .001 |
| Tempo | 120.07 | 104.93 | -0.47 | < .001 | .01 | < .001 |
| Liveness | .21 | .15 | -0.39 | < .001 | < .01 | < .001 |
| Speechiness | .1 | .07 | -0.35 | < .001 | < .01 | < .001 |

*Note.* For the statistically testing for equal means, we used Welch's *t*-test to correct for the unequal size and variance between the two datasets. All P-values were FDR-corrected. The following audio features were lower in the SPD compared to the MSSD: Loudness (Cohen's d = -1.25), Energy (Cohen's d = -1.46), Danceability (Cohen's d = -0.64), Valence (Cohen's d = -0.93), Tempo (Cohen's d = -0.47), Liveness (Cohen's d = -0.39), Speechiness (Cohen's d = -0.35). The following audio features were higher in the SPD compared to the MSSD: Acousticness (Cohen's d = 1.20), Instrumentalness (Cohen's d = 1.10). The linear discriminant analysis gives an indication of which audio features are important when separating the SPD and the MSSD dataset. The analysis was performed with weighted data due to the imbalanced between the MSSD and the SPD. The correct predictions reached 78.61% (MSSD = 79.86%, SPD = 77.36%). All P-values were FDR-corrected.

## Genre analysis

In order to paint a better picture of the music present in the Sleep Playlist Dataset, we reduced the many genres tags that each track is assigned by Spotify to one single genre from the list in Table 2. The most popular genre was *sleep*, followed by *pop*, *ambient* and *lo-fi* (Table 4). 45,993 tracks had unknown genres and 15,816 tracks were unable to be categorised and corresponded to 789 sub-genres. However, as the most prevalent uncategorised sub-genre only has 817 counts, none of the uncategorised sub-genres would appear in the top 20 genres present in the Sleep Playlist Dataset.

## Subgroup characteristics of sleep music

To assess the degree to which sleep music can be considered one homogeneous group of music or whether different subgroups exist within this category, we performed a *k*-means clustering analysis that revealed seven distinct clusters of tracks. We merged two of these clusters, as their mean tempi were close to multiples (140.6 BPM and 76.5 BPM) and their remaining audio features were highly similar. Tempo can be challenging to algorithmically determine and often half-time or double-time tempo is measured instead of the original tempo, so this occurrence is not surprising. Thus, our analysis revealed six musically meaningful subgroups of sleep music. Fig 3 illustrates the mean audio feature values for each subgroup in relation to the sample mean.

These clusters differ both in size and in the distribution of their audio features. To improve interpretability, we inspected the tracks included in the clusters, in order to assign each cluster a descriptive tag. The audio features of clusters 1, 2 and 3 are substantially different from the average, with low Instrumentalness, high Energy, high Tempo and high Loudness. Cluster 1 (N = 8,275) is characterised by high Speechiness, hence its name "Speechy Tracks". It contains mainly rap, R&B or lofi tracks (S3 Table). Cluster 2 (N = 30,959) and cluster 3 (N = 30,721) are similar in their audio feature distribution and the tracks that they contain. They mostly contain popular songs of the moment, pop and indie tracks with some lofi, and R&B tracks. The main difference between them is that cluster 3 has high Acousticness whereas cluster 2 does not (Fig

**Table 4. Number of occurrences of each of the top 20 genre categories in the Sleep Playlist Dataset.**

| Genre Category | Number of Occurrences |
| --- | --- |
| Sleep | 46,984 |
| Pop | 29,550 |
| Ambient | 13,660 |
| Lo-fi | 13,622 |
| Rap | 7,081 |
| Lullaby | 6,613 |
| Classical | 5,917 |
| Instrumental | 5,571 |
| Jazz | 5,187 |
| Meditation | 4,113 |
| Rock | 3,742 |
| Background music | 3,716 |
| Indie | 2,924 |
| Soundtrack | 2,768 |
| Folk | 2,066 |
| Country | 1,894 |
| R&B | 1,569 |
| World | 1,548 |
| Christian | 1,454 |
| New Age | 1,040 |

Note: House, Electronic, Metal, Alternative, Funk, Punk, Reggae, Bluegrass, Blues, Gospel and Opera had less than 1,000 counts.

3). Cluster 2 is therefore tagged "Radio Tracks", and cluster 3 is tagged "Acoustic Radio Tracks". Cluster 4 is the largest cluster (N = 117,237) and contains meditation tracks, healing music with nature sounds, continuous drone music or ambient music (S3 Table). Cluster 5 (N = 32,651) has higher Danceability than "Ambient Tracks" and the tracks from this cluster are mostly instrumental compositions, either piano cover tracks, classical or jazz instrumentals (S3 Table). By comparing tracks from cluster 5 and "Ambient Tracks", it is apparent that tracks from cluster 5 usually have a stable pulse, which is expected in instrumental compositions but which is often absent or less salient in "Ambient Tracks" due to the continuous and floating feel of ambient music. Cluster 5 is composed of non-ambient instrumental tracks, hence its tag "Instrumental Tracks". Cluster 6 (N = 5,783) is defined by a high Liveness value, and many of the tracks in this cluster is general pop or Christian tracks, the latter which has a tendency to be recorded live.

## Discussion

By building a large collection of musical tracks associated with sleep, we show that sleep music is characterised by lower Tempo, Loudness, Energy and Tempo and is more likely to have high Instrumentalness and Acousticness values than general music. However, even within sleep music, a large variation of music features remains. Our results show that sleep music can be divided into six distinct clusters, with half of the clusters mirroring the characteristics of sleep music overall and half having higher Energy and lower Instrumentalness.

Previously, studies have focused on the music genres used for sleep, and a British survey study found that classical music was the most frequent genre mentioned (32%), followed by

Subgroups of sleep music - audio features relative to average

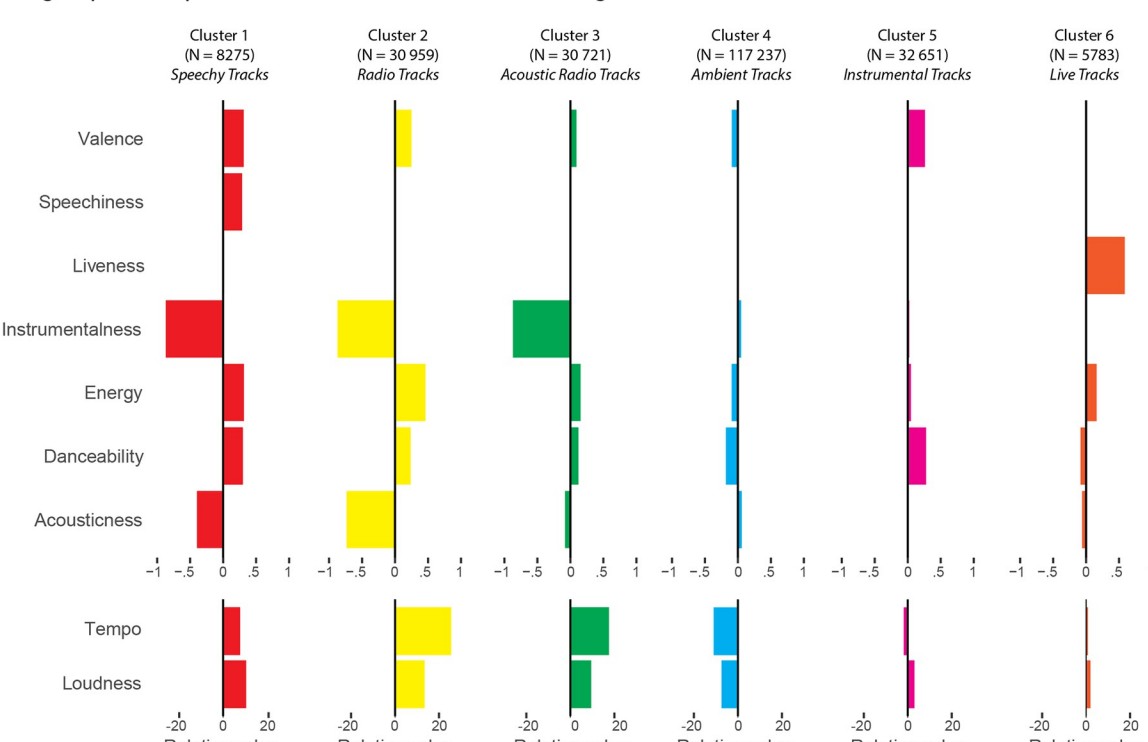

**Fig 3. Overview of subgroups of sleep music.** The six clusters' audio features are here shown in relation to the grand average value. A positive value indicates that the cluster is characterised by a relative increase in the audio feature's value, and a negative value indicates a relative decrease.

rock (11%), pop (8%), acoustic (7%), jazz (6%), soundtrack (6%) and ambient (6%) [17]. Similarly, an Australian survey study found that of the pieces of music participants rated as successfully helping them fall asleep, 18.5% were classical, 12.3% pop, 12.3% ambient, 10.8% folk and 10% alternative, with 11 different genres in total [21]. Interestingly, the most frequent genres in the current study were *sleep*, *pop*, *ambient* and *lo-fi*, and classical music was only the 7th most frequent. The incongruence of findings could be because both previous studies were based in one single country with a limited number of participants whereas this current study used a global approach. This being said, the studies agree that no single type of music is most listened to by the general population to fall asleep, accentuating the need for music-based sleep interventions to include many different choices of genres [17, 21].

In addition to genre characteristics, the present study adds to the current knowledge base by examining the audio features of the music. Overall, the characteristics of sleep music revealed by this study are in line with previous research on diurnal fluctuations in music listening behaviour. A recent study found that reduced tempo, loudness and energy was characteristic of music listened to during the night and the early morning [26]. Similarly, the average musical intensity has been found to decrease during the evening hours [27]. In addition, experimental research has highlighted the importance of low tempo and loudness for arousal reduction in response to music [37–39]. The importance of a slow tempo may be explained by the entrainment of autonomous biological oscillators such as respiration and heart rate to external stimuli like the beat of the music [40, 41]. Additionally, there is also evidence for neural entrainment to musical rhythms at both beat and meter frequencies [42, 43]. Thus, it could be

argued that music with a slow tempo may promote sleep by enhancing low-frequency activity in the brain [44].

Even though our findings provide evidence of the general soothing characteristics of sleep music, we also show that there is much more to sleep music than standard relaxation music. Our results reveal that the sleep associated music varies substantially with regard to the audio features and music characteristics. The large variation described above is accentuated by the six subgroups we identified based on their audio features. The largest subgroup of the Sleep Playlist Dataset was "Ambient music" which is the most expected type of music when looking at music used for relaxation as it has low Danceability and Energy, and high Instrumentalness and Acousticness. These represent the universal and predominant characteristics of music used for sleep. However, different combinations of audio features were found in the other sub-groups ("Acoustic Radio Tracks", "Radio Track", "Speechy Tracks"). Surprisingly, these sub-groups included popular contemporary tracks, which have high Energy and Danceability, and low Instrumentalness and Acousticness. For example, by counting the times a given track occur in the playlists included in our dataset, we find that the most popular track which appeared 245 times was "Dynamite" by the k-pop band *BTS*. This track does not match previous descriptions of relaxation music [18–20] and is instead an up-beat track filled with syncopated and groovy melodic hooks and a busy rhythm section. Other popular sleep tracks included "Jealous" by *Labrinth* or "lovely (with Khalid)" by *Billie Eilish* and *Khalid* that appeared 62 and 60 times respectively (S4 Table). Both these tracks are characterised by medium-low tempo (85 and 115 BPM respectively, yet with an emphasis on half-time (57.5 BPM) on the latter), and a sparse instrumentation with focus on long melodic lines.

One could argue that music with high Energy and Danceability would be counterproductive for relaxation and sleep, however it is possible that they could increase relaxation when considering the interplay between repeated exposure, familiarity and predictive processing. In short, predictive coding is a general theory of brain function which proposes that the brain continuously makes predictions about the world that are compared to sensory input, and if found wrong, triggers a prediction error signal used to refine future predictions [45, 46]. Hence, if music contains many surprising elements, this would lead to many prediction errors [47–50]. With repeated exposure, the brain gets increasingly precise at predicting the music. As a piece of music becomes increasingly familiar there is a corresponding decrease in attentional focus and in general energy use [51]. As such, it may be that familiar music even with high Energy and Danceability could facilitate relaxation due to its highly predictive nature. However, this relationship remains to be tested [47, 48, 52, 53]. Similarly, music that is very repetitive and constant over time might also result in increased relaxation due to familiarization with the piece and the increase of dynamic expectations [47]. In such a case, even music with, for example, high Tempo or high Energy might induce relaxation. However, the data presented here does not include track dynamics.

In addition to these surprising finds, more expected music is also present in our dataset. For instance, popular relaxation music pieces such as "Brahms Lullaby", "Clair de Lune" or "Canon in D" also appeared more than 100 times in the dataset as well as lullabies and nursery rhymes like "Twinkle Twinkle Little Star" and "Incy wincy spider". However, these were often present in various variations, with differing instrumentation, and hence different audio features. This is a weakness of using purely data-driven audio features to characterise music, as they are based on the recorded audio, and not on the notated music.

One explanation for the wide variety of tracks in our dataset could be the different motivations to listen to music before sleeping. Trahan et al. found four different reasons why people listen to music before bed: (1) in order to change their state (mental, physical, or relaxation), (2) to provide security, (3) as distraction, or (4) just by habit [17]. Certain types of music may

be more suitable than others depending on the reason for using music as sleep aid. For example, music that leads to relaxation is usually linked to slow Tempo, low Energy, and high Instrumentalness, such as the tracks that are within the "Ambient Tracks" and "Instrumental Tracks" subgroups. However, a different motivation for music use before sleep, such as mood regulation, might be done better with tracks that are already liked by the listener. Because the motivation of the listener might have a large influence on the type of music people choose to listen to before bed, future research should investigate to what extent different reasons for using music as sleep aid may drive the specific choice of music. Furthermore, research on music used for emotion regulation shows that people do not always choose the music that facilitates a positive effect [54, 55]. Therefore, future studies should clarify if the different subgroups of sleep music do promote sleep equally well while taking music preferences into account.

Overall, the results of this study clearly highlight the variation within sleep music, and the need to move beyond genre descriptions towards more specific analyses of the audio features of the music. These results can help inform the choice of music for clinical studies, music therapy or personal use. Previously, some clinical trials have used researcher-selected music [56] while others have given participants a choice among pre-selected playlists [57]. So far, it is not clear to which degree the choice of music has an impact on the effect on sleep [58, 59].

When considering the results of this study, it is worth taking into account the limitation that we do not have demographic information on the specific users of the sleep music. As such, we cannot exclude the possibility that the dataset might be skewed by a certain demographic, such as having more younger people or more of one gender. However, we know that there are Spotify users in 92 different countries, covering many continents [25]. Furthermore, it is known that streamers of online music cover all ages [23]. Therefore, we consider this the most global sleep music dataset to date, with the largest age-range and demographic variability.

In summary, our study used the digital traces of music streaming to shed light on the widespread human practice of using music as sleep aid. Poor sleep is a growing problem in society and our study contributes to this field, by providing new knowledge on both the universality and diversity of sleep music characteristics that can help inform future music-interventions as well as bringing us a step closer to understanding how music is used to regulate emotions and arousal by millions of people in everyday life.

## Supporting information

**S1 Table. Playlist exclusion criteria.**
(DOCX)

**S2 Table. Descriptive statistics of the Sleep Playlist Dataset.**
(DOCX)

**S3 Table. Most frequent musical genres, audio features and tracks based on trackID of the 6 clusters.**
(DOCX)

**S4 Table. Top 20 most frequent tracks based on trackID.**
(DOCX)

## Author Contributions

**Conceptualization:** Ole Adrian Heggli, Peter Vuust, Kira Vibe Jespersen.

**Data curation:** Rebecca Jane Scarratt, Ole Adrian Heggli.

**Formal analysis:** Rebecca Jane Scarratt, Ole Adrian Heggli.

**Investigation:** Rebecca Jane Scarratt, Kira Vibe Jespersen.

**Methodology:** Rebecca Jane Scarratt, Ole Adrian Heggli, Kira Vibe Jespersen.

**Project administration:** Peter Vuust, Kira Vibe Jespersen.

**Resources:** Peter Vuust.

**Software:** Ole Adrian Heggli.

**Supervision:** Ole Adrian Heggli, Peter Vuust, Kira Vibe Jespersen.

**Validation:** Rebecca Jane Scarratt, Ole Adrian Heggli.

**Visualization:** Rebecca Jane Scarratt, Ole Adrian Heggli.

**Writing – original draft:** Rebecca Jane Scarratt, Ole Adrian Heggli, Kira Vibe Jespersen.

**Writing – review & editing:** Ole Adrian Heggli, Peter Vuust, Kira Vibe Jespersen.

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
