## [Decision Letter · Decision Letter 0]

17 Oct 2022

PONE-D-22-01555The music that people use to sleep: universal and subgroup characteristicsPLOS ONE

Dear Dr. Jespersen,

Thank you for submitting your manuscript to PLOS ONE. After careful consideration, we feel that it has merit but does not fully meet PLOS ONE’s publication criteria as it currently stands. Therefore, we invite you to submit a revised version of the manuscript that addresses the points raised during the review process. Both reviewers suggested some major changes to improve the presentation. Please, take into account Reviewer 1's comment and clarify the precise subject of your paper more clearly in the outset.Please, consider suggestions of Reviewer 2 also carefully.

We look forward to receiving your revised manuscript.

Kind regards,

Gábor Vattay, PhD, DSc

Academic Editor

PLOS ONE

Journal Requirements:

1, Please ensure that your manuscript meets PLOS ONE's style requirements, including those for file naming. The PLOS ONE style templates can be found at

2. In your Methods section, please include additional information about your dataset and ensure that you have included a statement specifying whether the collection and analysis method complied with the terms and conditions for the source of the data.

“Center for Music in the Brain is funded by the Danish National Research Foundation (DNRF117). The authors received no specific funding for this work.”

Reviewers' comments:

Reviewer's Responses to Questions

**Comments to the Author**

1. Is the manuscript technically sound, and do the data support the conclusions?

Reviewer #1: Yes

Reviewer #2: Yes

2. Has the statistical analysis been performed appropriately and rigorously? 

Reviewer #1: Yes

Reviewer #2: Yes

3. Have the authors made all data underlying the findings in their manuscript fully available?

Reviewer #1: Yes

Reviewer #2: Yes

4. Is the manuscript presented in an intelligible fashion and written in standard English?

Reviewer #1: Yes

Reviewer #2: Yes

5. Review Comments to the Author

Reviewer #1: The title of the paper promises that the study is about music to be used to sleep.

Unfortunately this is not the case, because the study does not investigate whether the music really causes sleep. The study is just what people think that the music might help to sleep. This is a different question. No effects on sleep were investigated. It is just the assumption that people think that the music has a positive effect on sleep. In order to make this clear, I suggest to modify the title of the paper and to modify the abstract as well.

The processing of Spotify music tracks is nice and entertaining. The processing seems to be done in a correct way. However, all this misses the effect on sleep. It could be so, that you search for music which creates good mood or promisses love or peace and no assessment on what the music promisses or what the assumed effect is.

The paper is on the analysis of music tracks and this can be a valid analysis, but this is not my area of expertise.

Reviewer #2: Authors measured the characteristics of music associated with sleep by extracting audio features from a large number of tracks (N = 225,626) retrieved from sleep playlists at the global streaming platform Spotify. The results suggested that sleep music is characterised by lower Tempo, Loudness, Energy and Tempo and is more likely to have high Instrumentalness and Acousticness values than general music. However, even within sleep music, a large variation of music features remains. The large variation described by the six subgroups authors identified based on their audio features. As the authors stated the findings reveal previously unknown aspects of sleep music and highlight the individual variation in the choice of music for facilitating sleep.

In general, the article is well-written and the authors presented their results in a proper way. I have just some minor comments about the manuscript.

1. I would recommend to clarify the importance of the study and add some practical implication e.g. how music therapist get help to find music to their patients with sleep problems and/or how the findings can be helpful for researchers who are planning experimental studies to understand more about the connections between listening to music and sleep quality.

2. I suggest to write an independent section about the limitation of the study (maybe authors can include here what they mentioned in the supplementary discussion)

3. One specific comment about the novelty of the study, i.e. the results suggested that sleep music is characterised by lower Tempo, Loudness, Energy than general music but people also use music with high Energy, and Danceability would be counterproductive for relaxation and sleep. In one hand, I agree with the authors this type of genre of music could increase relaxation when considering the interplay between repeated exposure, familiarity and predictive processing. In other hand, not only high Energy and Danceability but faster tempo appeared also in the first three cluster can be counterproductive with relaxation effect and sleep according to the entrainment of autonomous biological oscillators such as respiration and heart rate to external stimuli like the beat of the music. I am not sure only familiarity of the music overcome all of these effects may increase physiological arousal however a music with fast tempo and highly repetitive rhythm does not vary throughout of the piece, may help to create a hypnotic feel in the listeners and it has a sleep inducing effect.

6. PLOS authors have the option to publish the peer review history of their article (what does this mean?). If published, this will include your full peer review and any attached files.

Reviewer #1: **Yes: **Thomas Penzel

Reviewer #2: **Yes: **László Harmat, Ph.D.

---

## [Author Response · Author response to Decision Letter 0]

9 Nov 2022

Thanks to editor and reviewers for their time and effort in reviewing our manuscript. We believe the review has helped clarify important aspects of the manuscript and improved the overall quality.

Journal Requirements:

Response: Thanks for the guidelines. We have edited the manuscript to ensure that it meets PLOS ONE’s style requirements.

2. In your Methods section, please include additional information about your dataset and ensure that you have included a statement specifying whether the collection and analysis method complied with the terms and conditions for the source of the data.

Response: Thank you. We have included a sentence stating that we did not collect personal information and that we comply with Spotify’s terms of use. Please see p. 6 ll. 119-121.

“Center for Music in the Brain is funded by the Danish National Research Foundation (DNRF117). The authors received no specific funding for this work.”

Response: Thanks for this clarification. We have removed the text from the Acknowledgements section and would like you to kindly update the Funding Statement to “Center for Music in the Brain is funded by the Danish National Research Foundation (DNRF117). The authors received no specific funding for this work.”. This is also included in the cover letter.

Reviewers' comments:

Reviewer #1: 

1. The title of the paper promises that the study is about music to be used to sleep. Unfortunately this is not the case, because the study does not investigate whether the music really causes sleep. The study is just what people think that the music might help to sleep. This is a different question. No effects on sleep were investigated. It is just the assumption that people think that the music has a positive effect on sleep. In order to make this clear, I suggest to modify the title of the paper and to modify the abstract as well.

The processing of Spotify music tracks is nice and entertaining. The processing seems to be done in a correct way. However, all this misses the effect on sleep. It could be so, that you search for music which creates good mood or promisses love or peace and no assessment on what the music promisses or what the assumed effect is. The paper is on the analysis of music tracks and this can be a valid analysis, but this is not my area of expertise.

Response: Thank you very much! We have changed the title of the manuscript to "The audio features of sleep music: universal and subgroup characteristics" and edited the abstract to ensure that they clearly reflect the focus of this study.

Reviewer #2: 

Authors measured the characteristics of music associated with sleep by extracting audio features from a large number of tracks (N = 225,626) retrieved from sleep playlists at the global streaming platform Spotify. The results suggested that sleep music is characterised by lower Tempo, Loudness, Energy and Tempo and is more likely to have high Instrumentalness and Acousticness values than general music. However, even within sleep music, a large variation of music features remains. The large variation described by the six subgroups authors identified based on their audio features. As the authors stated the findings reveal previously unknown aspects of sleep music and highlight the individual variation in the choice of music for facilitating sleep.

In general, the article is well-written and the authors presented their results in a proper way. I have just some minor comments about the manuscript.

1. I would recommend to clarify the importance of the study and add some practical implication e.g. how music therapist get help to find music to their patients with sleep problems and/or how the findings can be helpful for researchers who are planning experimental studies to understand more about the connections between listening to music and sleep quality.

Response: Thank you very much for these suggestions. In the introduction, we have clarified the contribution of this study, and in the discussion, we have added a section on the relevance of the study to both clinical practice as well as research in the field of music and sleep.

2. I suggest to write an independent section about the limitation of the study (maybe authors can include here what they mentioned in the supplementary discussion)

Response: Thank you. We have added a section on limitations implementing the considerations from the supplementary information.

3. One specific comment about the novelty of the study, i.e. the results suggested that sleep music is characterised by lower Tempo, Loudness, Energy than general music but people also use music with high Energy, and Danceability would be counterproductive for relaxation and sleep. In one hand, I agree with the authors this type of genre of music could increase relaxation when considering the interplay between repeated exposure, familiarity and predictive processing. In other hand, not only high Energy and Danceability but faster tempo appeared also in the first three cluster can be counterproductive with relaxation effect and sleep according to the entrainment of autonomous biological oscillators such as respiration and heart rate to external stimuli like the beat of the music. I am not sure only familiarity of the music overcome all of these effects may increase physiological arousal however a music with fast tempo and highly repetitive rhythm does not vary throughout of the piece, may help to create a hypnotic feel in the listeners, and it has a sleep inducing effect.

Response: Thank you for pointing this out. We agree that familiarity may not be the only factor to explain why people use faster tempo music for sleep, and we have expanded our discussion to consider also the dynamics of each track in terms of repetition and variation within the music track.

---

## [Decision Letter · Decision Letter 1]

28 Nov 2022

The audio features of sleep music: universal and subgroup characteristics

PONE-D-22-01555R1

Dear Dr. Jespersen,

We’re pleased to inform you that your manuscript has been judged scientifically suitable for publication and will be formally accepted for publication once it meets all outstanding technical requirements.

Kind regards,

Gábor Vattay, PhD, DSc

Academic Editor

PLOS ONE

Additional Editor Comments (optional):

Reviewers' comments:

Reviewer's Responses to Questions

**Comments to the Author**

1. If the authors have adequately addressed your comments raised in a previous round of review and you feel that this manuscript is now acceptable for publication, you may indicate that here to bypass the “Comments to the Author” section, enter your conflict of interest statement in the “Confidential to Editor” section, and submit your "Accept" recommendation.

Reviewer #1: All comments have been addressed

Reviewer #2: All comments have been addressed

2. Is the manuscript technically sound, and do the data support the conclusions?

Reviewer #1: Yes

Reviewer #2: Yes

3. Has the statistical analysis been performed appropriately and rigorously? 

Reviewer #1: Yes

Reviewer #2: Yes

4. Have the authors made all data underlying the findings in their manuscript fully available?

Reviewer #1: Yes

Reviewer #2: Yes

5. Is the manuscript presented in an intelligible fashion and written in standard English?

Reviewer #1: Yes

Reviewer #2: Yes

6. Review Comments to the Author

Reviewer #1: Thank you for addressing my concerns regarding the title. I am more confident with the new title. Also the remaining manuscript was improved.

Reviewer #2: Thank you for the author to the changes in the manuscript. I review this paper for acceptance to the editor.

7. PLOS authors have the option to publish the peer review history of their article (what does this mean?). If published, this will include your full peer review and any attached files.

Reviewer #1: No

Reviewer #2: **Yes: **László Harmat Ph.D

---

## [Editor Report · Acceptance letter]

1 Dec 2022

PONE-D-22-01555R1 

The audio features of sleep music: universal and subgroup characteristics 

Dear Dr. Jespersen:

I'm pleased to inform you that your manuscript has been deemed suitable for publication in PLOS ONE. Congratulations! Your manuscript is now with our production department. 

Kind regards, 

on behalf of

Dr. Gábor Vattay 

Academic Editor

PLOS ONE